# Collaborative Recurrent Autoencoder: Recommend while Learning to Fill in the Blanks

**Hao Wang, Xingjian Shi, Dit-Yan Yeung**
Hong Kong University of Science and Technology
{hwangaz,xshiab,dyyeung}@cse.ust.hk

## Abstract

Hybrid methods that utilize both content and rating information are commonly used in many recommender systems. However, most of them use either handcrafted features or the bag-of-words representation as a surrogate for the content information but they are neither effective nor natural enough. To address this problem, we develop a collaborative recurrent autoencoder (CRAE) which is a denoising recurrent autoencoder (DRAE) that models the generation of content sequences in the collaborative filtering (CF) setting. The model generalizes recent advances in recurrent deep learning from i.i.d. input to non-i.i.d. (CF-based) input and provides a new denoising scheme along with a novel learnable pooling scheme for the recurrent autoencoder. To do this, we first develop a hierarchical Bayesian model for the DRAE and then generalize it to the CF setting. The synergy between denoising and CF enables CRAE to make accurate recommendations while learning to fill in the blanks in sequences. Experiments on real-world datasets from different domains (CiteULike and Netflix) show that, by jointly modeling the order-aware generation of sequences for the content information and performing CF for the ratings, CRAE is able to significantly outperform the state of the art on both the recommendation task based on ratings and the sequence generation task based on content information.

## 1   Introduction

With the high prevalence and abundance of Internet services, recommender systems are becoming increasingly important to attract users because they can help users make effective use of the information available. Companies like Netflix have been using recommender systems extensively to target users and promote products. Existing methods for recommender systems can be roughly categorized into three classes [13]: content-based methods that use the user profiles or product descriptions only, collaborative filtering (CF) based methods that use the ratings only, and hybrid methods that make use of both. Hybrid methods using both types of information can get the best of both worlds and, as a result, usually outperform content-based and CF-based methods.

Among the hybrid methods, collaborative topic regression (CTR) [20] was proposed to integrate a topic model and probabilistic matrix factorization (PMF) [15]. CTR is an appealing method in that it produces both promising and interpretable results. However, CTR uses a bag-of-words representation and ignores the order of words and the local context around each word, which can provide valuable information when learning article representation and word embeddings. Deep learning models like convolutional neural networks (CNN) which use layers of sliding windows (kernels) have the potential of capturing the order and local context of words. However, the kernel size in a CNN is fixed during training. To achieve good enough performance, sometimes an ensemble of multiple CNNs with different kernel sizes has to be used. A more natural and adaptive way of modeling text sequences would be to use gated recurrent neural network (RNN) models [8, 3, 18]. A gated RNN takes in one

word (or multiple words) at a time and lets the learned gates decide whether to incorporate or to forget the word. Intuitively, if we can generalize gated RNNs to the CF setting (non-i.i.d.) to jointly model the generation of sequences and the relationship between items and users (rating matrices), the recommendation performance could be significantly boosted.

Nevertheless, very few attempts have been made to develop feedforward deep learning models for CF, let alone recurrent ones. This is due partially to the fact that deep learning models, like many machine learning models, assume i.i.d. inputs. [16, 6, 7] use restricted Boltzmann machines and RNN instead of the conventional matrix factorization (MF) formulation to perform CF. Although these methods involve both deep learning and CF, they actually belong to CF-based methods because they do not incorporate the content information like CTR, which is crucial for accurate recommendation. [14] uses low-rank MF in the last weight layer of a deep network to reduce the number of parameters, but it is for classification instead of recommendation tasks. There have also been nice explorations on music recommendation [10, 25] in which a CNN or deep belief network (DBN) is directly used for content-based recommendation. However, the models are deterministic and less robust since the noise is not explicitly modeled. Besides, the CNN is directly linked to the ratings making the performance suffer greatly when the ratings are sparse, as will be shown later in our experiments. Very recently, collaborative deep learning (CDL) [23] is proposed as a probabilistic model for joint learning of a probabilistic stacked denoising autoencoder (SDAE) [19] and collaborative filtering. However, CDL is a feedforward model that uses bag-of-words as input and it does not model the order-aware generation of sequences. Consequently, the model would have *inferior recommendation performance* and is *not capable of generating sequences* at all, which will be shown in our experiments. Besides order-awareness, another drawback of CDL is its *lack of robustness* (see Section 3.1 and 3.5 for details). To address these problems, we propose a hierarchical Bayesian generative model called collaborative recurrent autoencoder (CRAE) to jointly model the order-aware generation of sequences (in the content information) and the rating information in a CF setting. Our main contributions are:

- By exploiting recurrent deep learning collaboratively, CRAE is able to sophisticatedly model the generation of items (sequences) while extracting the implicit relationship between items (and users). We design a novel pooling scheme for pooling variable-length sequences into fixed-length vectors and also propose a new denoising scheme to effectively avoid overfitting. Besides for recommendation, CRAE can also be used to generate sequences on the fly.
- To the best of our knowledge, CRAE is the first model that bridges the gap between RNN and CF, especially with respect to hybrid methods for recommender systems. Besides, the Bayesian nature also enables CRAE to seamlessly incorporate other auxiliary information to further boost the performance.
- Extensive experiments on real-world datasets from different domains show that CRAE can substantially improve on the state of the art.

## 2   Problem Statement and Notation

Similar to [20], the recommendation task considered in this paper takes implicit feedback [9] as the training and test data. There are $J$ items (e.g., articles or movies) in the dataset. For item $j$, there is a corresponding sequence consisting of $T_j$ words where the vector $\mathbf{e}_t^{(j)}$ specifies the $t$-th word using the 1-of-$S$ representation, i.e., a vector of length $S$ with the value 1 in only one element corresponding to the word and 0 in all other elements. Here $S$ is the vocabulary size of the dataset. We define an $I$-by-$J$ binary rating matrix $\mathbf{R} = [\mathbf{R}_{ij}]_{I \times J}$ where $I$ denotes the number of users. For example, in the *CiteULike* dataset, $\mathbf{R}_{ij} = 1$ if user $i$ has article $j$ in his or her personal library and $\mathbf{R}_{ij} = 0$ otherwise. Given some of the ratings in $\mathbf{R}$ and the corresponding sequences of words $\mathbf{e}_t^{(j)}$ (e.g., titles of articles or plots of movies), the problem is to predict the other ratings in $\mathbf{R}$.

In the following sections, $\mathbf{e}_t'^{(j)}$ denotes the noise-corrupted version of $\mathbf{e}_t^{(j)}$ and $(\mathbf{h}_t^{(j)}; \mathbf{s}_t^{(j)})$ refers to the concatenation of the two $K_W$-dimensional column vectors. All input weights (like $\mathbf{Y}_e$ and $\mathbf{Y}_e^i$) and recurrent weights (like $\mathbf{W}_e$ and $\mathbf{W}_e^i$) are of dimensionality $K_W$-by-$K_W$. The output state $\mathbf{h}_t^{(j)}$, gate units (e.g., $\mathbf{h}_t^{o(j)}$), and cell state $\mathbf{s}_t^{(j)}$ are of dimensionality $K_W$. $K$ is the dimensionality of the final representation $\boldsymbol{\gamma}_j$, middle-layer units $\boldsymbol{\theta}_j$, and latent vectors $\mathbf{v}_j$ and $\mathbf{u}_i$. $\mathbf{I}_K$ or $\mathbf{I}_{K_W}$ denotes a $K$-by-$K$ or $K_W$-by-$K_W$ identity matrix. For convenience we use $\mathbf{W}^+$ to denote the collection of all weights and biases. Similarly $\mathbf{h}_t^+$ is used to denote the collection of $\mathbf{h}_t$, $\mathbf{h}_t^i$, $\mathbf{h}_t^f$, and $\mathbf{h}_t^o$.

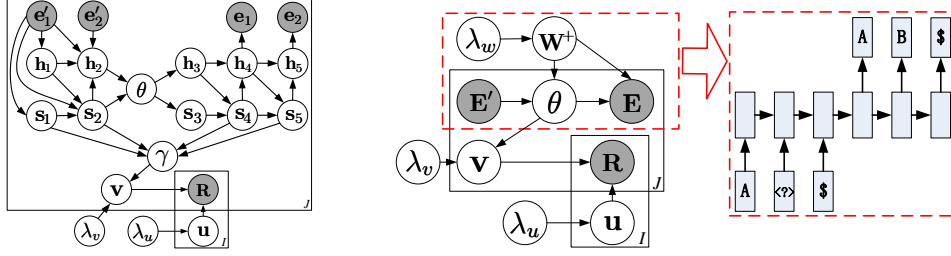

Figure 1: On the left is the graphical model for an example CRAE where $T_j = 2$ for all $j$. To prevent clutter, the hyperparameters for beta-pooling, all weights, biases, and links between $\mathbf{h}_t$ and $\boldsymbol{\gamma}$ are omitted. On the right is the graphical model for the degenerated CRAE. An example recurrent autoencoder with $T_j = 3$ is shown. '$\langle ? \rangle$' is the $\langle$wildcard$\rangle$ and '\$' marks the end of a sentence. $\mathbf{E}'$ and $\mathbf{E}$ are used in place of $[\mathbf{e}_t'^{(j)}]_{t=1}^{T_j}$ and $[\mathbf{e}_t^{(j)}]_{t=1}^{T_j}$ respectively.

# 3    Collaborative Recurrent Autoencoder

In this section we will first propose a generalization of the RNN called *robust recurrent networks* (RRN), followed by the introduction of two key concepts, *wildcard denoising* and *beta-pooling*, in our model. After that, the generative process of CRAE is provided to show how to generalize the RRN as a hierarchical Bayesian model from an i.i.d. setting to a CF (non-i.i.d.) setting.

## 3.1    Robust Recurrent Networks

One problem with RNN models like long short-term memory networks (LSTM) is that the computation is deterministic without taking the noise into account, which means it is not robust especially with insufficient training data. To address this *robustness* problem, we propose RRN as a type of noisy gated RNN. In RRN, the gates and other latent variables are designed to incorporate noise, making the model more robust. Note that unlike [4, 5], the noise in RRN is directly propagated back and forth in the network, without the need for using separate neural networks to approximate the distributions of the latent variables. This is much more efficient and easier to implement. Here we provide the generative process of RRN. Using $t = 1 \ldots T_j$ to index the words in the sequence, we have (we drop the index $j$ for items for notational simplicity):

$$\mathbf{x}_{t-1} \sim \mathcal{N}(\mathbf{W}_w \mathbf{e}_{t-1}, \lambda_s^{-1} \mathbf{I}_{K_W}), \quad \mathbf{a}_{t-1} \sim \mathcal{N}(\mathbf{Y}\mathbf{x}_{t-1} + \mathbf{W}\mathbf{h}_{t-1} + \mathbf{b}, \lambda_s^{-1} \mathbf{I}_{K_W}) \tag{1}$$

$$\mathbf{s}_t \sim \mathcal{N}(\sigma(\mathbf{h}_{t-1}^f) \odot \mathbf{s}_{t-1} + \sigma(\mathbf{h}_{t-1}^i) \odot \sigma(\mathbf{a}_{t-1}), \lambda_s^{-1} \mathbf{I}_{K_W}), \tag{2}$$

where $\mathbf{x}_t$ is the word embedding of the $t$-th word, $\mathbf{W}_w$ is a $K_W$-by-$S$ word embedding matrix, $\mathbf{e}_t$ is the 1-of-$S$ representation mentioned above, $\odot$ stands for the element-wise product operation between two vectors, $\sigma(\cdot)$ denotes the sigmoid function, $\mathbf{s}_t$ is the cell state of the $t$-th word, and $\mathbf{b}$, $\mathbf{Y}$, and $\mathbf{W}$ denote the biases, input weights, and recurrent weights respectively. The forget gate units $\mathbf{h}_t^f$ and the input gate units $\mathbf{h}_t^i$ in Equation (2) are drawn from Gaussian distributions depending on their corresponding weights and biases $\mathbf{Y}^f$, $\mathbf{W}^f$, $\mathbf{Y}^i$, $\mathbf{W}^i$, $\mathbf{b}^f$, and $\mathbf{b}^i$:

$$\mathbf{h}_t^f \sim \mathcal{N}(\mathbf{Y}^f \mathbf{x}_t + \mathbf{W}^f \mathbf{h}_t + \mathbf{b}^f, \lambda_s^{-1} \mathbf{I}_{K_W}), \quad \mathbf{h}_t^i \sim \mathcal{N}(\mathbf{Y}^i \mathbf{x}_t + \mathbf{W}^i \mathbf{h}_t + \mathbf{b}^i, \lambda_s^{-1} \mathbf{I}_{K_W}).$$

The output $\mathbf{h}_t$ depends on the output gate $\mathbf{h}_t^o$ which has its own weights and biases $\mathbf{Y}^o$, $\mathbf{W}^o$, and $\mathbf{b}^o$:

$$\mathbf{h}_t^o \sim \mathcal{N}(\mathbf{Y}^o \mathbf{x}_t + \mathbf{W}^o \mathbf{h}_t + \mathbf{b}^o, \lambda_s^{-1} \mathbf{I}_{K_W}), \quad \mathbf{h}_t \sim \mathcal{N}(\tanh(\mathbf{s}_t) \odot \sigma(\mathbf{h}_{t-1}^o), \lambda_s^{-1} \mathbf{I}_{K_W}). \tag{3}$$

In the RRN, information of the processed sequence is contained in the cell states $\mathbf{s}_t$ and the output states $\mathbf{h}_t$, both of which are column vectors of length $K_W$. Note that RRN can be seen as a generalized and Bayesian version of LSTM [1]. Similar to [18, 3], two RRNs can be concatenated to form an encoder-decoder architecture.

## 3.2    Wildcard Denoising

Since the input and output are identical here, unlike [18, 3] where the input is from the source language and the output is from the target language, this naive RRN autoencoder can suffer from serious overfitting, even after taking noise into account and reversing sequence order (we find that

reversing sequence order in the decoder [18] does not improve the recommendation performance). One natural way of handling it is to borrow ideas from the *denoising autoencoder* [19] by randomly dropping some of the words in the encoder. Unfortunately, directly dropping words may mislead the learning of transition between words. For example, if we drop the word 'is' in the sentence 'this is a good idea', the encoder will wrongly learn the subsequence 'this a', which never appears in a grammatically correct sentence. Here we propose another denoising scheme, called *wildcard denoising*, where a special word '⟨wildcard⟩' is added to the vocabulary and we randomly select some of the words and replace them with '⟨wildcard⟩'. This way, the encoder RRN will take 'this ⟨wildcard⟩ a good idea' as input and successfully avoid learning wrong subsequences. We call this *denoising recurrent autoencoder* (DRAE). Note that the word '⟨wildcard⟩' also has a corresponding word embedding. Intuitively this wildcard denoising RRN autoencoder learns to *fill in the blanks* in sentences automatically. We find this denoising scheme much better than the naive one. For example, in dataset *CiteULike* wildcard denoising can provide a relative accuracy boost of about 20%.

### 3.3 Beta-Pooling

The RRN autoencoders would produce a representation vector for each input word. In order to facilitate the factorization of the rating matrix, we need to pool the sequence of vectors into one single vector of fixed length $2K_W$ before it is further encoded into a $K$-dimensional vector. A natural way is to use a weighted average of the vectors. Unfortunately different sequences may need weights of *different size*. For example, pooling a sequence of $8$ vectors needs a weight vector with $8$ entries while pooling a sequence of $50$ vectors needs one with $50$ entries. In other words, we need a weight vector of *variable length* for our pooling scheme. To tackle this problem, we propose to use a beta distribution. If six vectors are to be pooled into one single vector (using weighted average), we can use the area $w_p$ in the range $(\frac{p-1}{6}, \frac{p}{6})$ of the $x$-axis of the probability density function (PDF) for the beta distribution $\text{Beta}(a,b)$ as the pooling weight. Then the resulting pooling weight vector becomes $\mathbf{y} = (w_1, \ldots, w_6)^T$. Since the total area is always $1$ and the $x$-axis is bounded, the beta distribution is perfect for this type of variable-length pooling (hence the name *beta-pooling*). If we set the hyperparameters $a = b = 1$, it will be equivalent to average pooling. If $a$ is set large enough and $b > a$ the PDF will peak slightly to the left of $x = 0.5$, which means that the last time step of the encoder RRN is directly used as the pooling result. With only two parameters, beta-pooling is able to pool vectors flexibly enough without having the risk of overfitting the data.

### 3.4 CRAE as a Hierarchical Bayesian Model

Following the notation in Section 2 and using the DRAE in Section 3.2 as a component, we then provide the generative process of the CRAE (note that $t$ indexes words or time steps, $j$ indexes sentences or documents, and $T_j$ is the number of words in document $j$):

**Encoding** ($t = 1, 2, \ldots, T_j$): Generate $\mathbf{x}'^{(j)}_{t-1}$, $\mathbf{a}^{(j)}_{t-1}$, and $\mathbf{s}^{(j)}_t$ according to Equation (1)-(2).

**Compression and decompression** ($t = T_j + 1$):

$$\boldsymbol{\theta}_j \sim \mathcal{N}(\mathbf{W}_1(\mathbf{h}^{(j)}_{T_j}; \mathbf{s}^{(j)}_{T_j}) + \mathbf{b}_1, \lambda_s^{-1}\mathbf{I}_K), \quad (\mathbf{h}^{(j)}_{T_j+1}; \mathbf{s}^{(j)}_{T_j+1}) \sim \mathcal{N}(\mathbf{W}_2 \tanh(\boldsymbol{\theta}_j) + \mathbf{b}_2, \lambda_s^{-1}\mathbf{I}_{2K_W}). \quad (4)$$

**Decoding** ($t = T_j + 2, T_j + 3, \ldots, 2T_j + 1$): Generate $\mathbf{a}^{(j)}_{t-1}$, $\mathbf{s}^{(j)}_t$, and $\mathbf{h}^{(j)}_t$ according to Equation (1)-(3), after which generate:

$$\mathbf{e}^{(j)}_{t-T_j-2} \sim \text{Mult}(\text{softmax}(\mathbf{W}_g \mathbf{h}^{(j)}_t + \mathbf{b}_g)).$$

**Beta-pooling and recommendation**:

$$\boldsymbol{\gamma}_j \sim \mathcal{N}(\tanh(\mathbf{W}_1 f_{a,b}(\{(\mathbf{h}^{(j)}_t; \mathbf{s}^{(j)}_t)\}_t) + \mathbf{b}_1), \lambda_s^{-1}\mathbf{I}_K) \quad (5)$$

$$\mathbf{v}_j \sim \mathcal{N}(\boldsymbol{\gamma}_j, \lambda_v^{-1}\mathbf{I}_K), \quad \mathbf{u}_i \sim \mathcal{N}(\mathbf{0}, \lambda_u^{-1}\mathbf{I}_K), \quad \mathbf{R}_{ij} \sim \mathcal{N}(\mathbf{u}_i^T \mathbf{v}_j, \mathbf{C}_{ij}^{-1}).$$

Note that each column of the weights and biases in $\mathbf{W}^+$ is drawn from $\mathcal{N}(\mathbf{0}, \lambda_w^{-1}\mathbf{I}_{K_W})$ or $\mathcal{N}(\mathbf{0}, \lambda_w^{-1}\mathbf{I}_K)$. In the generative process above, the input gate $\mathbf{h}^i_{t-1}{}^{(j)}$ and the forget gate $\mathbf{h}^f_{t-1}{}^{(j)}$ can be drawn as described in Section 3.1. $\mathbf{e}'^{(j)}_t$ denotes the corrupted word (with the embedding

$\mathbf{x}'^{(j)}_t$) and $\mathbf{e}^{(j)}_t$ denotes the original word (with the embedding $\mathbf{x}^{(j)}_t$). $\lambda_w$, $\lambda_u$, $\lambda_s$, and $\lambda_v$ are hyperparameters and $\mathbf{C}_{ij}$ is a confidence parameter ($\mathbf{C}_{ij} = \alpha$ if $\mathbf{R}_{ij} = 1$ and $\mathbf{C}_{ij} = \beta$ otherwise). Note that if $\lambda_s$ goes to infinity, the Gaussian distribution (e.g., in Equation (4)) will become a Dirac delta distribution centered at the mean. The compression and decompression act like a bottleneck between two Bayesian RRNs. The purpose is to reduce overfitting, provide necessary nonlinear transformation, and perform dimensionality reduction to obtain a more compact final representation $\boldsymbol{\gamma}_j$ for CF. The graphical model for an example CRAE where $T_j = 2$ for all $j$ is shown in Figure 1(left). $f_{a,b}(\{(\mathbf{h}^{(j)}_t; \mathbf{s}^{(j)}_t)\}_t)$ in Equation (5) is the result of beta-pooling with hyperparameters $a$ and $b$. If we denote the cumulative distribution function of the beta distribution as $F(x; a, b)$, $\boldsymbol{\phi}^{(j)}_t = (\mathbf{h}^{(j)}_t; \mathbf{s}^{(j)}_t)$ for $t = 1, \ldots, T_j$, and $\boldsymbol{\phi}^{(j)}_t = (\mathbf{h}^{(j)}_{t+1}; \mathbf{s}^{(j)}_{t+1})$ for $t = T_j + 1, \ldots, 2T_j$, then we have $f_{a,b}(\{(\mathbf{h}^{(j)}_t; \mathbf{s}^{(j)}_t)\}_t) = \sum_{t=1}^{2T_j}(F(\frac{t}{2T_j}, a, b) - F(\frac{t-1}{2T_j}, a, b))\boldsymbol{\phi}_t$. Please see Section 3 of the supplementary materials for details (including hyperparameter learning) of beta-pooling. From the generative process, we can see that both CRAE and CDL are Bayesian deep learning (BDL) models (as described in [24]) with a perception component (DRAE in CRAE) and a task-specific component.

## 3.5 Learning

According to the CRAE model above, all parameters like $\mathbf{h}^{(j)}_t$ and $\mathbf{v}_j$ can be treated as random variables so that a full Bayesian treatment such as methods based on variational approximation can be used. However, due to the extreme nonlinearity and the CF setting, this kind of treatment is non-trivial. Besides, with CDL [23] and CTR [20] as our primary baselines, it would be fairer to use maximum a posteriori (MAP) estimates, which is what CDL and CTR do.

**End-to-end joint learning**: Maximization of the posterior probability is equivalent to maximizing the joint log-likelihood of $\{\mathbf{u}_i\}$, $\{\mathbf{v}_j\}$, $\mathbf{W}^+$, $\{\boldsymbol{\theta}_j\}$, $\{\boldsymbol{\gamma}_j\}$, $\{\mathbf{e}^{(j)}_t\}$, $\{\mathbf{e}'^{(j)}_t\}$, $\{\mathbf{h}^{+(j)}_t\}$, $\{\mathbf{s}^{(j)}_t\}$, and $\mathbf{R}$ given $\lambda_u$, $\lambda_v$, $\lambda_w$, and $\lambda_s$:

$$\mathscr{L} = \log p(\mathrm{DRAE}|\lambda_s, \lambda_w) - \frac{\lambda_u}{2}\sum_i \|\mathbf{u}_i\|_2^2 - \frac{\lambda_v}{2}\sum_j \|\mathbf{v}_j - \boldsymbol{\gamma}_j\|_2^2 - \sum_{i,j}\frac{\mathbf{C}_{ij}}{2}(\mathbf{R}_{ij} - \mathbf{u}_i^T\mathbf{v}_j)^2$$

$$- \frac{\lambda_s}{2}\sum_j \|\tanh(\mathbf{W}_1 f_{a,b}(\{(\mathbf{h}^{(j)}_t; \mathbf{s}^{(j)}_t)\}_t) + \mathbf{b}_1) - \boldsymbol{\gamma}_j\|_2^2,$$

where $\log p(\mathrm{DRAE}|\lambda_s, \lambda_w)$ corresponds to the prior and likelihood terms for DRAE (including the encoding, compression, decompression, and decoding in Section 3.4) involving $\mathbf{W}^+$, $\{\boldsymbol{\theta}_j\}$, $\{\mathbf{e}^{(j)}_t\}$, $\{\mathbf{e}'^{(j)}_t\}$, $\{\mathbf{h}^{+(j)}_t\}$, and $\{\mathbf{s}^{(j)}_t\}$. For simplicity and computational efficiency, we can fix the hyperparameters of beta-pooling so that $\mathrm{Beta}(a, b)$ peaks slightly to the left of $x = 0.5$ (e.g., $a = 9.8 \times 10^7$, $b = 1 \times 10^8$), which leads to $\boldsymbol{\gamma}_j = \tanh(\boldsymbol{\theta}_j)$ (a treatment for the more general case with learnable $a$ or $b$ is provided in the supplementary materials). Further, if $\lambda_s$ approaches infinity, the terms with $\lambda_s$ in $\log p(\mathrm{DRAE}|\lambda_s, \lambda_w)$ will vanish and $\boldsymbol{\gamma}_j$ will become $\tanh(\mathbf{W}_1(\mathbf{h}^{(j)}_{T_j}, \mathbf{s}^{(j)}_{T_j}) + \mathbf{b}_1)$. Figure 1(right) shows the graphical model of a degenerated CRAE when $\lambda_s$ approaches positive infinity and $b > a$ (with very large $a$ and $b$). Learning this degenerated version of CRAE is equivalent to jointly training a wildcard denoising RRN and an encoding RRN coupled with the rating matrix. If $\lambda_v \ll 1$, CRAE will further degenerate to a two-step model where the representation $\boldsymbol{\theta}_j$ learned by the DRAE is directly used for CF. On the contrary if $\lambda_v \gg 1$, the decoder RRN essentially vanishes. Both extreme cases can greatly degrade the predictive performance, as shown in the experiments.

**Robust nonlinearity on distributions**: Different from [23, 22], nonlinear transformation is performed *after* adding the noise with precision $\lambda_s$ (e.g. $\mathbf{a}^{(j)}_t$ in Equation (1)). In this case, the input of the nonlinear transformation is a *distribution* rather than a deterministic *value*, making the nonlinearity more robust than in [23, 22] and leading to more efficient and direct learning algorithms than CDL.

Consider a univariate Gaussian distribution $\mathcal{N}(x|\mu, \lambda_s^{-1})$ and the sigmoid function $\sigma(x) = \frac{1}{1+\exp(-x)}$, the expectation (see Section 6 of the supplementary materials for details):

$$E(x) = \int \mathcal{N}(x|\mu, \lambda_s^{-1})\sigma(x)dx = \sigma(\kappa(\lambda_s)\mu), \tag{6}$$

Equation (6) holds because the convolution of a sigmoid function with a Gaussian distribution can be approximated by another sigmoid function. Similarly, we can approximate $\sigma(x)^2$ with $\sigma(\rho_1(x + \rho_0))$,

where $\rho_1 = 4 - 2\sqrt{2}$ and $\rho_0 = -\log(\sqrt{2}+1)$. Hence the variance

$$D(x) \approx \int \mathcal{N}(x|\mu, \lambda_s^{-1}) \circ \Phi(\xi\rho_1(x+\rho_0))dx - E(x)^2 = \sigma\left(\frac{\rho_1(\mu+\rho_0)}{(1+\xi^2\rho_1^2\lambda_s^{-1})^{1/2}}\right) - E(x)^2 \approx \lambda_s^{-1}, \quad (7)$$

where we use $\lambda_s^{-1}$ to approximate $D(x)$ for computational efficiency. Using Equation (6) and (7), the Gaussian distribution in Equation (2) can be computed as:

$$\mathcal{N}(\sigma(\mathbf{h}_{t-1}^f) \odot \mathbf{s}_{t-1} + \sigma(\mathbf{h}_{t-1}^i) \odot \sigma(\mathbf{a}_{t-1}), \lambda_s^{-1}\mathbf{I}_{K_W})$$
$$\approx \mathcal{N}(\sigma(\kappa(\lambda_s)\bar{\mathbf{h}}_{t-1}^f) \odot \bar{\mathbf{s}}_{t-1} + \sigma(\kappa(\lambda_s)\bar{\mathbf{h}}_{t-1}^i) \odot \sigma(\kappa(\lambda_s)\bar{\mathbf{a}}_{t-1}), \lambda_s^{-1}\mathbf{I}_{K_W}), \quad (8)$$

where the superscript $(j)$ is dropped. We use overlines (e.g., $\bar{\mathbf{a}}_{t-1} = \mathbf{Y}_e\mathbf{x}_{t-1} + \mathbf{W}_e\mathbf{h}_{t-1} + \mathbf{b}_e$) to denote the mean of the distribution from which a hidden variable is drawn. By applying Equation (8) recursively, we can compute $\bar{\mathbf{s}}_t$ for any $t$. Similar approximation is used for $\tanh(x)$ in Equation (3) since $\tanh(x) = 2\sigma(2x) - 1$. This way the feedforward computation of DRAE would be *seamlessly chained* together, leading to more efficient learning algorithms than the layer-wise algorithms in [23, 22] (see Section 6 of the supplementary materials for more details).

**Learning parameters**: To learn $\mathbf{u}_i$ and $\mathbf{v}_j$, block coordinate ascent can be used. Given the current $\mathbf{W}^+$, we can compute $\boldsymbol{\gamma}$ as $\boldsymbol{\gamma} = \tanh(\mathbf{W}_1 f_{a,b}(\{(\mathbf{h}_t^{(j)}; \mathbf{s}_t^{(j)})\}_t) + \mathbf{b}_1)$ and get the following update rules:

$$\mathbf{u}_i \leftarrow (\mathbf{V}\mathbf{C}_i\mathbf{V}^T + \lambda_u\mathbf{I}_K)^{-1}\mathbf{V}\mathbf{C}_i\mathbf{R}_i$$
$$\mathbf{v}_j \leftarrow (\mathbf{U}\mathbf{C}_i\mathbf{U}^T + \lambda_v\mathbf{I}_K)^{-1}(\mathbf{U}\mathbf{C}_j\mathbf{R}_j + \lambda_v\tanh(\mathbf{W}_1 f_{a,b}(\{(\mathbf{h}_t^{(j)}; \mathbf{s}_t^{(j)})\}_t) + \mathbf{b}_1)^T),$$

where $\mathbf{U} = (\mathbf{u}_i)_{i=1}^I$, $\mathbf{V} = (\mathbf{v}_j)_{j=1}^J$, $\mathbf{C}_i = \mathrm{diag}(\mathbf{C}_{i1}, \ldots, \mathbf{C}_{iJ})$ is a diagonal matrix, and $\mathbf{R}_i = (\mathbf{R}_{i1}, \ldots, \mathbf{R}_{iJ})^T$ is a column vector containing all the ratings of user $i$.

Given $\mathbf{U}$ and $\mathbf{V}$, $\mathbf{W}^+$ can be learned using the back-propagation algorithm according to Equation (6)-(8) and the generative process in Section 3.4. Alternating the update of $\mathbf{U}$, $\mathbf{V}$, and $\mathbf{W}^+$ gives a local optimum of $\mathscr{L}$. After $\mathbf{U}$ and $\mathbf{V}$ are learned, we can predict the ratings as $\mathbf{R}_{ij} = \mathbf{u}_i^T\mathbf{v}_j$.

## 4 Experiments

In this section, we report some experiments on real-world datasets from different domains to evaluate the capabilities of recommendation and automatic generation of missing sequences.

### 4.1 Datasets

We use two datasets from different real-world domains. *CiteULike* is from [20] with $5,551$ users and $16,980$ items (articles with text). *Netflix* consists of $407,261$ users, $9,228$ movies, and $15,348,808$ ratings after removing users with less than 3 positive ratings (following [23], ratings larger than 3 are regarded as positive ratings). Please see Section 7 of the supplementary materials for details.

### 4.2 Evaluation Schemes

**Recommendation**: For the recommendation task, similar to [21, 23], $P$ items associated with each user are randomly selected to form the training set and the rest is used as the test set. We evaluate the models when the ratings are in different degrees of density ($P \in \{1, 2, \ldots, 5\}$). For each value of $P$, we repeat the evaluation five times with different training sets and report the average performance.

Following [20, 21], we use recall as the performance measure since the ratings are in the form of implicit feedback [9, 12]. Specifically, a zero entry may be due to the fact that the user is not interested in the item, or that the user is not aware of its existence. Thus precision is not a suitable performance measure. We sort the predicted ratings of the candidate items and recommend the top $M$ items for the target user. The recall@$M$ for each user is then defined as:

$$\text{recall@}M = \frac{\text{\# items that the user likes among the top } M}{\text{\# items that the user likes}}.$$

The average recall over all users is reported.

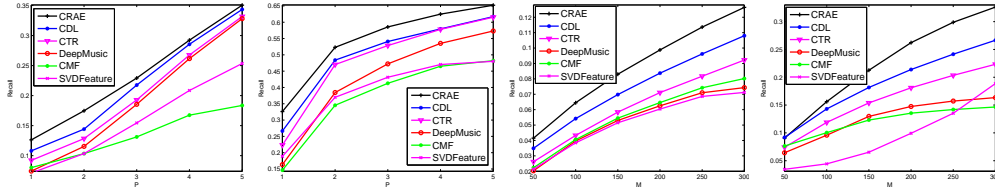

Figure 2: Performance comparison of CRAE, CDL, CTR, DeepMusic, CMF, and SVDFeature based on recall@$M$ for datasets *CiteULike* and *Netflix*. $P$ is varied from 1 to 5 in the first two figures.

We also use another evaluation metric, mean average precision (mAP), in the experiments. Exactly the same as [10], the cutoff point is set at 500 for each user.

**Sequence generation on the fly**: For the sequence generation task, we set $P = 5$. In terms of content information (e.g., movie plots), we randomly select 80% of the items to include their content in the training set. The trained models are then used to predict (generate) the content sequences for the other 20% items. The BLEU score [11] is used to evaluate the quality of generation. To compute the BLEU score in *CiteULike* we use the titles as training sentences (sequences). Both the titles and sentences in the abstracts of the articles (items) are used as reference sentences. For *Netflix*, the first sentences of the plots are used as training sentences. The movie names and sentences in the plots are used as reference sentences. A higher BLEU score indicates higher quality of sequence generation. Since CDL, CTR, and PMF *cannot generate sequences directly*, a nearest neighborhood based approach is used with the resulting $\mathbf{v}_j$. Note that this task is extremely difficult because the sequences of the test set are *unknown during both the training and testing phases*. For this reason, this task is impossible for existing machine translation models like [18, 3].

## 4.3 Baselines and Experimental Settings

The models for comparison are listed as follows:

- **CMF**: Collective Matrix Factorization [17] is a model incorporating different sources of information by simultaneously factorizing multiple matrices.
- **SVDFeature**: SVDFeature [2] is a model for feature-based collaborative filtering. In this paper we use the bag-of-words as raw features to feed into SVDFeature.
- **DeepMusic**: DeepMusic [10] is a feedforward model for music recommendation mentioned in Section 1. We use the best performing variant as our baseline.
- **CTR**: Collaborative Topic Regression [20] is a model performing topic modeling and collaborative filtering simultaneously as mentioned in the previous section.
- **CDL**: Collaborative Deep Learning (CDL) [23] is proposed as a probabilistic feedforward model for joint learning of a probabilistic SDAE [19] and CF.
- **CRAE**: Collaborative Recurrent Autoencoder is our proposed *recurrent* model. It jointly performs collaborative filtering and learns the generation of content (sequences).

In the experiments, we use 5-fold cross validation to find the optimal hyperparameters for CRAE and the baselines. For CRAE, we set $\alpha = 1$, $\beta = 0.01$, $K = 50$, $K_W = 100$. The wildcard denoising rate is set to $0.4$. See Section 5.1 of the supplementary materials for details.

## 4.4 Quantitative Comparison

**Recommendation**: The first two plots of Figure 2 show the recall@$M$ for the two datasets when $P$ is varied from 1 to 5. As we can see, CTR outperforms the other baselines except for CDL. Note that as previously mentioned, in both datasets DeepMusic suffers badly from overfitting when the rating matrix is extremely sparse ($P = 1$) and achieves comparable performance with CTR when the rating matrix is dense ($P = 5$). CDL as the strongest baseline consistently outperforms other baselines. By jointly learning the order-aware generation of content (sequences) and performing collaborative filtering, CRAE is able to outperform all the baselines by a margin of $0.7\% \sim 1.9\%$ (a relative boost of $2.0\% \sim 16.7\%$) in *CiteULike* and $3.5\% \sim 6.0\%$ (a relative boost of $5.7\% \sim 22.5\%$) in *Netflix*. Note that since the standard deviation is minimal ($3.38 \times 10^{-5} \sim 2.56 \times 10^{-3}$), it is not included in the figures and tables to avoid clutter.

The last two plots of Figure 2 show the recall@$M$ for *CiteULike* and *Netflix* when $M$ varies from 50 to 300 and $P = 1$. As shown in the plots, the performance of DeepMusic, CMF, and SVDFeature is

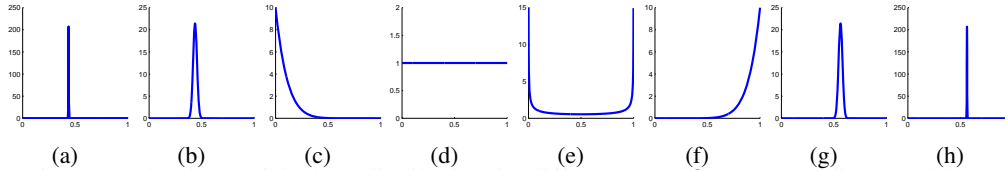

(a)  (b)  (c)  (d)  (e)  (f)  (g)  (h)

Figure 3: The shape of the beta distribution for different $a$ and $b$ (corresponding to Table 1).

Table 1: Recall@300 for beta-pooling with different hyperparameters

| a | 31112 | 311 | 1 | 1 | 0.4 | 10 | 400 | 40000 |
|---|---|---|---|---|---|---|---|---|
| b | 40000 | 400 | 10 | 1 | 0.4 | 1 | 311 | 31112 |
| Recall | 12.17 | 12.54 | 10.48 | 11.62 | 11.08 | 10.72 | **12.71** | 12.22 |

Table 2: mAP for two datasets

|  | CRAE | CDL | CTR | DeepMusic | CMF | SVDFeature |
|---|---|---|---|---|---|---|
| *CiteULike* | **0.0123** | 0.0091 | 0.0071 | 0.0058 | 0.0061 | 0.0056 |
| *Netflix* | **0.0301** | 0.0275 | 0.0211 | 0.0156 | 0.0144 | 0.0173 |

Table 3: BLEU score for two datasets

|  | CRAE | CDL | CTR | PMF |
|---|---|---|---|---|
| *CiteULike* | **46.60** | 21.14 | 31.47 | 17.85 |
| *Netflix* | **48.69** | 6.90 | 17.17 | 11.74 |

similar in this setting. Again CRAE is able to outperform the baselines by a large margin and the margin gets larger with the increase of $M$.

As shown in Figure 3 and Table 1, we also investigate the effect of $a$ and $b$ in beta-pooling and find that in DRAE: (1) temporal average pooling performs poorly ($a = b = 1$); (2) most information concentrates near the bottleneck; (3) the right of the bottleneck contains more information than the left. Please see Section 4 of the supplementary materials for more details.

As another evaluation metric, Table 2 compares different models based on mAP. As we can see, compared with CDL, CRAE can provide a relative boost of $35\%$ and $10\%$ for *CiteULike* and *Netflix*, respectively. Besides quantitative comparison, **qualitative comparison** of CRAE and CDL is provided in Section 2 of the supplementary materials. In terms of time cost, CDL needs 200 epochs (40s/epoch) while CRAE needs about 80 epochs (150s/epoch) for optimal performance.

**Sequence generation on the fly**: To evaluate the ability of sequence generation, we compute the BLEU score of the sequences (titles for *CiteULike* and plots for *Netflix*) generated by different models. As mentioned in Section 4.2, this task is impossible for existing machine translation models like [18, 3] due to the lack of source sequences. As we can see in Table 3, CRAE achieves a BLEU score of $46.60$ for *CiteULike* and $48.69$ for *Netflix*, which is much higher than CDL, CTR and PMF. Incorporating the content information when learning user and item latent vectors, CTR is able to outperform other baselines and CRAE can further boost the BLEU score by sophisticatedly and jointly modeling the generation of sequences and ratings. Note that although CDL is able to outperform other baselines in the recommendation task, it performs poorly when generating sequences on the fly, which demonstrates the importance of modeling each sequence recurrently as a whole rather than as separate words.

## 5 Conclusions and Future Work

We develop a collaborative recurrent autoencoder which can sophisticatedly model the generation of item sequences while extracting the implicit relationship between items (and users). We design a new pooling scheme for pooling variable-length sequences and propose a wildcard denoising scheme to effectively avoid overfitting. To the best of our knowledge, CRAE is the first model to bridge the gap between RNN and CF. Extensive experiments show that CRAE can significantly outperform the state-of-the-art methods on both the recommendation and sequence generation tasks.

With its Bayesian nature, CRAE can easily be generalized to seamlessly incorporate auxiliary information (e.g., the citation network for *CiteULike* and the co-director network for *Netflix*) for further accuracy boost. Moreover, multiple Bayesian recurrent layers may be stacked together to increase its representation power. Besides making recommendations and guessing sequences on the fly, the wildcard denoising recurrent autoencoder also has potential to solve other challenging problems such as recovering the blurred words in ancient documents.

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
