[Supplementary Material]

# Supplementary Materials
# for Collaborative Recurrent Autoencoder

**Hao Wang, Xingjian Shi, Dit-Yan Yeung**
Hong Kong University of Science and Technology
{hwangaz,xshiab,dyyeung}@cse.ust.hk

## 1   Learning Beta-Pooling

As mentioned in the paper, $f_{a,b}(\{(\mathbf{h}_t^{(j)};\mathbf{s}_t^{(j)})\}_t)$ is the result of beta-pooling. The cumulative distribution function of the beta distribution $F(x;a,b) = \frac{B(x;a,b)}{B(a,b)} = I_x(a,b)$, where $B(x;a,b) = \int_0^x t^{a-1}(1-t)^{b-1}dt$ is the incomplete beta function and the denominator $B(a,b) = \frac{\Gamma(a+b)}{\Gamma(a)\,\Gamma(b)}$. $\Gamma(\cdot)$ is the gamma function and $I_x(a,b)$ is also called the regularized incomplete beta function. If we denote $\boldsymbol{\phi}_t^{(j)} = (\mathbf{h}_t^{(j)};\mathbf{s}_t^{(j)})$ for $t = 1,\ldots,T_j$ and $\boldsymbol{\phi}_t^{(j)} = (\mathbf{h}_{t+1}^{(j)};\mathbf{s}_{t+1}^{(j)})$ for $t = T_j+1,\ldots,2T_j$, we have $f_{a,b}(\{(\mathbf{h}_t^{(j)};\mathbf{s}_t^{(j)})\}_t) = \sum_{t=1}^{2T_j}(I_{\frac{t}{2T_j}}(a,b) - I_{\frac{t-1}{2T_j}}(a,b))\boldsymbol{\phi}_t$. Written this way, we can evaluate the gradient of $\mathscr{L}$ with respect to $a$ and $b$ and use gradient-based methods to learn them. To illustrate it more clearly, if we take $\lambda_s$ to positive infinity, fix $b = 1$ and try to learn the optimal value of $a$, we can maximize the following joint log-likelihood:

$$\mathscr{L} = -\sum_{i,j}\frac{\mathbf{C}_{ij}}{2}(\mathbf{R}_{ij} - \mathbf{u}_i^T\mathbf{v}_j)^2 - \frac{\lambda_v}{2}\sum_j \|\mathbf{v}_j - \tanh(\mathbf{W}_1\sum_{t=1}^{2T_j}[I_{\frac{t}{2T_j}}(a,1) - I_{\frac{t-1}{2T_j}}(a,1)]\boldsymbol{\phi}_t + \mathbf{b}_1)\|_2^2$$
$$+ \sum_j \sum_{t=T_j+2}^{2T_j+1} H(\mathbf{e}_{t-T_j-1}^{(j)}, \text{softmax}(\mathbf{W}_g\mathbf{h}_t^{(j)} + \mathbf{b}_g)) - \frac{\lambda_u}{2}\sum_i \|\mathbf{u}_i\|_2^2 - \frac{\lambda_w}{2}g(\mathbf{W}^+).$$

Note that $H(\cdot,\cdot)$ denotes the cross-entropy loss for generating words from $\text{Mult}(\text{softmax}(\mathbf{W}_g\mathbf{h}_t^{(j)} + \mathbf{b}_g))$. The term $-\frac{\lambda_w}{2}g(\mathbf{W}^+)$ corresponds to the prior of all weights and biases. Using the property of the regularized incomplete beta function that $I_x(a,1) = x^a$, the joint log-likelihood can be simplified to

$$\mathscr{L} = -\frac{\lambda_v}{2}\sum_j \|\mathbf{v}_j - \tanh(\mathbf{W}_1\sum_{t=1}^{2T_j}[(\frac{t}{2T_j})^a - (\frac{t-1}{2T_j})^a]\boldsymbol{\phi}_t + \mathbf{b}_1)\|_2^2 - \frac{\lambda_u}{2}\sum_i \|\mathbf{u}_i\|_2^2$$
$$- \sum_{i,j}\frac{\mathbf{C}_{ij}}{2}(\mathbf{R}_{ij} - \mathbf{u}_i^T\mathbf{v}_j)^2 + \sum_j \sum_{t=T_j+2}^{2T_j+1} H(\mathbf{e}_{t-T_j-1}^{(j)}, \text{softmax}(\mathbf{W}_g\mathbf{h}_t^{(j)} + \mathbf{b}_g)) - \frac{\lambda_w}{2}g(\mathbf{W}^+),$$

where $a$ only appears in the exponents of $(\frac{t}{2T_j})^a$ and $(\frac{t-1}{2T_j})^a$, which means we can easily get the gradient of $\mathscr{L}$ with respect to $a$ using the chain rule. After each epoch or minibatch, $a$ can be updated based on the gradient with the same learning rate.

## 2   Qualitative Comparison

In order to gain a better insight into CRAE, we train CRAE and CDL in the sparsest setting ($P = 1$) with dataset *CiteULike* and use them to recommend articles for two example users. The corresponding articles for the target users in the training set and the top 10 recommended articles are shown in

Table 1: Qualitative comparison between CRAE and CDL

| the rated article | Bayesian adaptive user profiling with explicit and implicit feedback | |
|---|---|---|
| | User I (CRAE) | in user's lib? |
| top 10 articles | 1. Incorporating user search behavior into relevance feedback | no |
| | **2. Query chains: learning to rank from implicit feedback** | **yes** |
| | **3. Implicit feedback for inferring user preference: a bibliography** | **yes** |
| | 4. Modeling user rating profiles for collaborative filtering | no |
| | 5. Improving retrieval performance by relevance feedback | no |
| | 6. Language models for relevance feedback | no |
| | **7. Context-sensitive information retrieval using implicit feedback** | **yes** |
| | **8. Implicit user modeling for personalized search** | **yes** |
| | **9. Model-based feedback in the language modeling approach to information retrieval** | **yes** |
| | **10. User language model for collaborative personalized search** | **yes** |
| | User I (CDL) | in user's lib? |
| top 10 articles | **1. Implicit feedback for inferring user preference: a bibliography** | **yes** |
| | 2. Seeing stars: Exploiting class relationships for sentiment categorization with respect to rating scales | no |
| | 3. A knowledge-based approach for interpreting genome-wide expression profiles | no |
| | 4. A tutorial on particle filters for online non-linear/non-gaussian Bayesian tracking | no |
| | **5. Query chains: learning to rank from implicit feedback** | **yes** |
| | 6. Mapreduce: simplified data processing on large clusters | no |
| | 7. Correlating user profiles from multiple folksonomies | no |
| | 8. Evolving object-oriented designs with refactorings | no |
| | 9. Trapping of neutral sodium atoms with radiation pressure | no |
| | 10. A scheme for efficient quantum computation with linear optics | no |
| the rated article | Taxonomy of trust: categorizing P2P reputation systems | |
| | User II (CRAE) | in user's lib? |
| top 10 articles | 1. Effects of positive reputation systems | no |
| | **2. Trust in recommender systems** | **yes** |
| | 3. trust metrics in recommender systems | no |
| | 4. The Structure of Collaborative Tagging Systems | no |
| | 5. Effects of energy policies on industry expansion in renewable energy | no |
| | **6. Limited reputation sharing in P2P systems** | **yes** |
| | 7. Survey of wireless indoor positioning techniques and systems | no |
| | 8. Design coordination in distributed environments using virtual reality systems | no |
| | **9. Propagation of trust and distrust** | **yes** |
| | 10. Physiological measures of presence in stressful virtual environments | no |
| | User II (CDL) | in user's lib? |
| top 10 articles | **1. Trust in recommender systems** | **yes** |
| | 2. Position Paper, Tagging, Taxonomy, Flickr, Article, ToRead | no |
| | 3. A taxonomy of workflow management systems for grid computing | no |
| | 4. Usage patterns of collaborative tagging systems | no |
| | 5. Semantic blogging and decentralized knowledge management | no |
| | 6. Flickr tag recommendation based on collective knowledge | no |
| | 7. Delivering real-world ubiquitous location systems | no |
| | 8. Shilling recommender systems for fun and profit | no |
| | 9. Privacy risks in recommender systems | no |
| | 10. Probabilistic reasoning in intelligent systems networks of plausible inference | no |

Table 1. Note that in the sparsest setting the recommendation task is extremely challenging since there is only one single article for each user in the training set.

As we can see, CRAE successfully identified User I as a researcher working on **information retrieval** with interest in **user modeling using user feedback**. Consequently, CRAE achieves a high precision of $60\%$ by focusing its recommendations on articles about information retrieval, user modeling, and relevance feedback. On the other hand, the topics of articles recommended by CDL span from **visual tracking** (Article 4) to **bioinformatics** (Article 3) and **programming language** (Article 8). One possible reason is that CDL uses the bag-of-words representation as input and consider each word separately without taking into account the local context of words. For example, looking into CDL's recommendations more closely, we can find that Article 3 (on bioinformatics) and Article 4 (on visual tracking) are actually irrelevant to the training article '**Bayesian** adaptive user **profiling** with explicit and implicit feedback'. CDL probably recommends Article 3 because the word '**profiles**' in the title overlaps with the article in the training set. The same thing happens for Article 4 with a word '**Bayesian**'. With the recurrent learning in CRAE, a sequence is modeled as a whole instead of separate words. As a result, with the local context of each word taken into consideration, CRAE can recognize the whole phrase '**user profiling**', rather than 'user' or 'profiling', as a theme of the article.

A similar phenomenon is found for User II with the article 'Taxonomy of trust: categorizing P2P reputation **systems**'. CDL's recommendations bet on the single word 'systems' while CRAE identified the article to be on **trust propagation** from the words 'trust' and 'P2P'. In the end, CRAE achieves a precision of $30\%$ and CDL's precision is $10\%$.

Figure 1: The shape of the beta distribution for different $a$ and $b$ (corresponding to Table 2).

Table 2: Recall@300 for beta-pooling with different hyperparameters

| a | 31112 | 311 | 1 | 1 | 0.4 | 10 | 400 | 40000 |
|---|---|---|---|---|---|---|---|---|
| b | 40000 | 400 | 10 | 1 | 0.4 | 1 | 311 | 31112 |
| Recall | 12.17 | 12.54 | 10.48 | 11.62 | 11.08 | 10.72 | **12.71** | 12.22 |

## 3  Motivation of Beta-Pooling

The function $f_{a,b}(\{(\mathbf{h}_t^{(j)}; \mathbf{s}_t^{(j)})\}_t)$ is to pool the output states $\mathbf{h}_t^{(j)}$ and the cell states $\mathbf{s}_t^{(j)}$ of $2T_j$ steps (a $2K_W$-by-$2T_j$ matrix) into a single vector of length $2K_W$. If we denote the cumulative distribution function of the beta distribution as $F(x; a, b)$, $\boldsymbol{\phi}_t^{(j)} = (\mathbf{h}_t^{(j)}; \mathbf{s}_t^{(j)})$ for $t = 1, \ldots, T_j$, and $\boldsymbol{\phi}_t^{(j)} = (\mathbf{h}_{t+1}^{(j)}; \mathbf{s}_{t+1}^{(j)})$ for $t = T_j + 1, \ldots, 2T_j$, then we have

$$f_{a,b}(\{(\mathbf{h}_t^{(j)}; \mathbf{s}_t^{(j)})\}_t) = \sum_{t=1}^{2T_j}(F(\frac{t}{2T_j}, a, b) - F(\frac{t-1}{2T_j}, a, b))\boldsymbol{\phi}_t.$$

Note that $a$ and $b$ are hyperparameters here. In a generalized setting, they can be learned automatically. Essentially the motivation of beta-pooling is to *handle the variable length for different sequences using one unified distribution*.

When $a = 2$ and $b = 3$, the beta-pooling is close to average pooling but with larger weights to the left of the center (the bottleneck). Following the generative process, the output $\mathbf{h}_t^{(j)}$ and cell states $\mathbf{s}_t^{(j)}$ of each word are concatenated into $(\mathbf{h}_t^{(j)}; \mathbf{s}_t^{(j)})$. For each sequence, $(\mathbf{h}_t^{(j)}; \mathbf{s}_t^{(j)})$ of all timesteps are beta-pooled into a vector of length $2K_W$. The vector is then further encoded into the vector $\boldsymbol{\gamma}_j$ of length $K$, which is used to guide the CF for the rating matrix. Since the information flows in both ways, the rating matrix can, in return, provide useful information when the wildcard denoising recurrent autoencoder tries to learn to fill in the blanks. This two-way interaction enables both tasks (recommendation task and sequence generation task) to benefit from each other and results in more effective representation $\boldsymbol{\theta}_j$ for each item.

Note that the compression layer and the beta-pooling share the same weights and biases. If the hyperparameters of beta-pooling are fixed so that $\text{Beta}(a, b)$ peaks slightly to the left of $x = 0.5$, the generation of $\boldsymbol{\gamma}_j$ in the generative process is equivalent to directly setting $\boldsymbol{\gamma}_j = \tanh(\boldsymbol{\theta}_j)$ where $\boldsymbol{\theta}_j$ is the compressed representation we get from the compression layer. For example, $\text{Beta}(a, b)$ peaks slightly to the left of $x = 0.5$ (near $x = \frac{7}{16}$) when $a = 7778$, $b = 10000$, and $T_j = 4$. The only time step that interacts with the rating matrix is the one when $t = 4$, which is encoded into $\boldsymbol{\theta}_j$ and connected to the item latent vector $\mathbf{v}_j$.

## 4  Experiments on Beta-Pooling and Wildcard Denoising

As mentioned in the paper, beta-pooling is able to pool a sequence of $2T_j$ vectors into one single vector of the same size. Note that $T_j$ here can vary for different $j$. Hyperparameters $a$ and $b$ control the behavior of beta-pooling. When $a = b = 1$, beta-pooling is equivalent to temporal average pooling that takes the average of the $2T_j$ vectors. In an extreme case, $a$ and $b$ can be set such that the pooling result is equal to one of the $2T_j$ vectors (e.g., the $T_j$-th vector). Figure 1 shows the shape of the beta distribution for different $a$ and $b$. Table 2 shows the corresponding recall for different beta distributions in *CiteULike*. As we can see, the average pooling in Figure 1(d) and the pooling with an inverted bell curve in Figure 1(e) perform poorly. On the other hand, distributions in Figure 1(a), (b), (g), and (h) yield the highest accuracy, which means most information concentrates near the

Figure 2: The recall@$M$ for different $\lambda_v$.

bottleneck (middle) of DRAE. Among them, the distributions in Figure 1(b) and (g) outperform those in Figure 1(a) and (h). This shows that simply setting the pooling result to be the middle vector is not good enough and an aggregation of vectors near the middle would be a better choice. Comparing distributions in Figure 1(b) and (g), it can be seen that the latter slightly outperforms the former, probably because there are no input words in the decoder part of DRAE (as shown in the graphical model of CRAE), which makes the hidden and cell states in the decoder part more representative. Similar phenomena happen for Figure 1(a), (c), (f), and (h).

Note that since CRAE is a joint model, the information flows both ways through beta-pooling. For example, when $a = 400$ and $b = 311$, the item representations used for recommendation mostly come from the cell and output states near the bottleneck and in return, the rating information affects the learning of DRAE mainly through the cell and output states near the bottleneck.

As mentioned in the paper, for the wildcard denoising scheme, we find that in *CiteULike*, CRAE performs best with a wildcard denoising rate of $0.4$, achieving a recall@300 of $12.71\%$ while the number for CRAE with conventional denoising [3] (dropping words completely) is $10.53\%$ (slightly better than CDL). For reference, the recall of CRAE without any denoising is $9.14\%$. Similar phenomena are found in *Netflix*.

Note that DRAE is a much more general model than RNN autoencoders like [2, 1]. We also try reversing the order of each sequence in the decoder RNN as in [2, 1], but the performance only changes slightly.

## 5 Hyperparameters

We provide more details on the hyperparameters in this section.

### 5.1 Hyperparameter Settings

The vocabulary size $S$ (with the word $\langle wildcard \rangle$ included) is 15,050 and 17,949 for *CiteULike* and *Netflix* respectively. For CMF and SVDFeature, optimal regularization hyperparameters are used for different $P$. The learning rate is set to $0.005$ for SVDFeature. For DeepMusic, we find that the best performance is achieved using a CNN with two convolutional layers. For CTR, we find that it can achieve good prediction performance when $\lambda_u = 0.1$, $\lambda_v = 10$, and $K = 50$. For CDL, we use similar hyperparameters as mentioned in [5]. The denoising rate is set to $0.3$. Dropout rate, $\lambda_u$, $\lambda_v$, and $\lambda_n$ are set using the validation sets. For the sequence generation task, we postprocess the generated sequences by deleting consecutive repeated words (e.g., the word 'like' in the sentence 'I like like this idea'), as often done in RNN-based sentence generation models.

### 5.2 Hyperparameter Sensitivity

Figure 2 shows the recall@$M$ for *CiteULike* when $\lambda_v$ is from $0.001$ to $10$ ($P = 5$). As mentioned in the paper, when $\lambda_v \ll 1$ CRAE degenerates to a two-step model with no joint learning on the content sequences and ratings. If $\lambda_v \gg 1$ the decoder side of CRAE will essentially vanish. Apparently the performance suffers a lot in both extremes, which shows the effectiveness of joint learning in the full CRAE model.

## 6 Robust Nonlinearity on Distributions

Different from [5], nonlinear transformation is performed *after* adding the noise with precision $\lambda_s$. In this case, the input of the nonlinear transformation is a *distribution* rather than a deterministic *value*, making the nonlinearity more robust than in [5] and leading to more efficient and direct learning algorithms than CDL.

Consider a univariate Gaussian distribution $\mathcal{N}(x|\mu, \lambda_s^{-1})$ and the sigmoid function $\sigma(x) = \frac{1}{1+\exp(-x)}$, the expectation:

$$
\begin{aligned}
E(x) &= \int \mathcal{N}(x|\mu, \lambda_s^{-1})\sigma(x)dx \\
&\approx \int \mathcal{N}(x|\mu, \lambda_s^{-1})\Phi(\xi x)dx \\
&= \Phi(\xi \kappa(\lambda_s)\mu) = \sigma(\kappa(\lambda_s)\mu),
\end{aligned}
\tag{1}
$$

where the probit function $\Phi(x) = \int_{-\infty}^{x} \mathcal{N}(\theta|0,1)d\theta$, $\kappa(\lambda_s) = (1 + \xi^2\lambda_s^{-1})^{-\frac{1}{2}}$, and $\Phi(\xi x)$, with $\xi^2 = \frac{\pi}{8}$, is to approximate $\sigma(x)$ by matching the slope at the origin. Equation (1) holds because the convolution of a probit function with a Gaussian distribution is another probit function. Similarly, we can approximate $\sigma(x)^2$ with $\sigma(\rho_1(x + \rho_0))$ by matching both the value and the slope at the origin, where $\rho_1 = 4 - 2\sqrt{2}$ and $\rho_0 = -\log(\sqrt{2}+1)$. Hence the variance

$$
\begin{aligned}
D(x) &\approx \int \mathcal{N}(x|\mu, \lambda_s^{-1}) \circ \Phi(\xi\rho_1(x + \rho_0))dx - E(x)^2 \\
&= \sigma\left(\frac{\rho_1(\mu + \rho_0)}{(1 + \xi^2\rho_1^2\lambda_s^{-1})^{1/2}}\right) - E(x)^2 \approx \lambda_s^{-1},
\end{aligned}
\tag{2}
$$

where we use $\lambda_s^{-1}$ to approximate $D(x)$ for computational efficiency. Using Equation (1) and (2), the Gaussian distribution in for generating $\mathbf{s}_t$ can be computed as:

$$
\begin{aligned}
&\mathcal{N}(\sigma(\mathbf{h}_{t-1}^f) \odot \mathbf{s}_{t-1} + \sigma(\mathbf{h}_{t-1}^i) \odot \sigma(\mathbf{a}_{t-1}), \lambda_s^{-1}\mathbf{I}_{K_W}) \\
&\approx \mathcal{N}(\sigma(\kappa(\lambda_s)\overline{\mathbf{h}}_{t-1}^f) \odot \overline{\mathbf{s}}_{t-1} + \sigma(\kappa(\lambda_s)\overline{\mathbf{h}}_{t-1}^i) \odot \sigma(\kappa(\lambda_s)\overline{\mathbf{a}}_{t-1}), \lambda_s^{-1}\mathbf{I}_{K_W}),
\end{aligned}
\tag{3}
$$

where the superscript $(j)$ is dropped for clarity. We use overlines (e.g., $\overline{\mathbf{a}}_{t-1} = \mathbf{Y}_e\mathbf{x}_{t-1} + \mathbf{W}_e\mathbf{h}_{t-1} + \mathbf{b}_e$) to denote the mean of the distribution from which a hidden variable is drawn. By applying Equation (3) recursively, we can compute $\overline{\mathbf{s}}_t$ for any $t$. Similarly, since $\tanh(x) = 2\sigma(2x) - 1$, we have:

$$
\begin{aligned}
E(x) &= \int \mathcal{N}(x|\mu, \lambda_s^{-1})\tanh(x)dx \\
&\approx 2\sigma(x(0.25 + \xi^2\lambda_s^{-1})^{-\frac{1}{2}}) - 1,
\end{aligned}
\tag{4}
$$

which could be used to approximate $\mathbf{h}_t^{(j)} \sim \mathcal{N}(\tanh(\mathbf{s}_t^{(j)}) \odot \sigma(\mathbf{h}_{t-1}^o{}^{(j)}), \lambda_s^{-1}\mathbf{I}_{K_W})$. This way the feedforward computation of DRAE would be *seamlessly chained* together, leading to more efficient learning algorithms than the layer-wise algorithms in [5].

## 7 Datasets

We use two datasets from different real-world domains, one from CiteULike [1] and the other from Netflix. The first dataset, *CiteULike*, is from [4] with $5,551$ users and $16,980$ items (articles). The titles of the articles are used as content information (sequences of words) in our model. The second dataset, *Netflix*, consists of both movie ratings from the users and the plots (content information) for the movies. After removing users with less than 3 positive ratings (following [5], ratings larger than 3 are regarded as positive ratings) and movies without plots, we get $407,261$ users, $9,228$ movies, and $15,348,808$ ratings in the final dataset.

## Footnotes

[1]CiteULike allows users to create their own collections of articles. There are abstract, title, and tags for each article. More details about the CiteULike data can be found at `http://www.citeulike.org`.