[Reviews · NeurIPS 2016]

Reviewer 1

Summary

This paper aims to address recommendation with both collaborative ratings and content of items by deep learning. The content sequences are modeled by the proposed recurrent autoencoder and the learned representation is used to constrain item vectors for rating matrix factorization.

Qualitative Assessment

This paper improves the hybrid recommendation problem by using deep learning. The key novelty lies in that it models content information naturally as word sequences and a novel collaborative recurrent autoencoder is proposed. The paper is well written. In experiments, the authors show the superiority over several state-of-art deep learning methods for recommendation. I have some minor questions/suggestions: (1) Fig.1 is too far from the corresponding description in the main text. I suggest move it to P5. (2) When you model documents (rather than sentences), how do you handle sentence boundaries? It is better to discuss about this a bit. (3) For the decoder part, there is no x_t input. How to compute s_t and h_t in this case (i.e. line 157-158)? (4) Line 194: should it be "if \lambda_v \gg 1, tanh(\theta_j) is directly used for CF. When \lambda_v \ll 1, the rating factorization and deep representation learning is nearly decoupled"? (5) In Eqs(6) and (7), E(x) should be E(\sigma(x)), D(x) should be D(\sigma(x)). (6) It would be better if the sequence generation on the fly experiment provides some generated examples.

Confidence in this Review

2-Confident (read it all; understood it all reasonably well)


Reviewer 2

Summary

The paper proposes a novel recurrent autoencoder (CRAE) that learns to recommend e.g. movies to users on Netflix as well as generating word sequences. CRAE is particularly interesting because it models the noise distribution within the network state and variables (in the style of bayesian LSTM). Other novelties include the introduction of wildcard denoising and beta-pooling which make a lot of sense and prevent the system from overfitting e.g. learning to copy the sequences.

Qualitative Assessment

The idea of using wildcards makes a lot of sense to me. Reminds me of zeroing out the input images as done in "Deep Tracking" with Convolutional LSTMs. Please see if this is relevant to cite. I'd like a bit of clarification on beta pooling and why only beta distribution was chosen. I'd also recommend cutting out certain bits of the text to have a figure explaining the beta pooling properly. You don't seem to mention anything about gradient clipping etc. if you have used it. All normal distributions seem to have the same variance, please explain why? (e.g. Eq 1-2-3 etc.) The results seem to show that CRAE outperforms all the other methods. Different experiments in Table3 highlight how the denoising and beta-pooling helps - I think this clearly shows the benefits of their novel approach. typos: line 288: "deleting" instead of "delelting"

Confidence in this Review

2-Confident (read it all; understood it all reasonably well)


Reviewer 3

Summary

This paper proposes a joint model of sequential content signals (such as movie plot text) and user ratings. A stochastic recurrent neural network (RNN) is used to model the former, then producing a latent representation v that serves as the item component in the factorization of the ratings matrix. Several auxiliary techniques are proposed to improve performance: the aforementioned noise in the RNN, a dropout-like technique called 'wildcard denoising', and a Beta pooling operation. Experiments against five baseline models are performed on the CiteULike and Netflix datasets for recommendation and sequence generation tasks. The CRAE shows superior to competitive recall at k and superior BLEU scores.

Qualitative Assessment

Making recommendations by exploiting the plethora of text data that accompanies products seems like an under-explored area, especially using recent advances in deep language models. I commend the authors for contributing to this research direction. The CRAE seems to work well (at least in recall at k), performing at a high level on two real-world datasets. However, I think this paper would be a better fit for a more applied conference, such as KDD or RecSys, because there is little novelty to the model's core components. I'll address each individually, in order of (my perceived) importance: 1) Robust Recurrent Networks (RRN): The proposed RRN uses distributional activations that are backpropagated through directly. If this development was indeed as novel as the paper seems to suggest, then my rating would have been higher (probably "poster level" in the 'novelty' category). However, these exact techniques are used in "Fast Dropout Training" by S. Wang and C. Manning: http://nlp.stanford.edu/pubs/sidaw13fast.pdf This paper is not cited but absolutely should be. The RRN is simply a recurrent version of the Wang-Manning model. Furthermore, considering the RNN from Wang & Manning's asymptotic dropout perspective, I'm not so sure the hidden activations should be drawn from normal distributions since traditional feedforward dropout does not work well in RNNs. Alternatives preserving activations over time seem to work better (see https://arxiv.org/abs/1606.01305 and http://arxiv.org/abs/1512.05287). 2) "CRAE as a Hierarchical Bayesian Model": Despite the RRN's lack of novelty, Bayesian learning of a model as complex as the CRAE on datasets as large as Netflix would have been a laudable contribution. However, only MAP estimation is performed, thus relegating the priors to regularization terms. It's unclear how Bayesian learning of this model could be done at any realistic scale (which the authors admit, line 178) and so I don't think it's accurate to call it a Bayesian model (for example, one wouldn't call a neural network with L2 regularization a Bayesian neural network). 3) Wildcard denoising: Sampling a 'wildcard' word does seem to be a novel regularization technique that others may find useful. Although, the development itself is not novel enough to significantly increase the paper's score. 4) Beta pooling: The authors suggest pooling latent representations via partitioning the Beta distribution and weighting each vector by the corresponding area under the pdf. This would be interesting if the Beta parameters were learned (which they can be, as the supplementary materials discuss), but they seem to be set a priori in the experiments (from line 186). When the hyperparameters are fixed, the method reduces to weighted average pooling (I think?). This is not significantly new enough to warrant an increase from 'sub-standard'. Next, I'll move on to address the experiments. The authors test the CRAE for making recommendations using recall at K, saying precision at K "is not suitable" "due to the fact that [a zero entry may denote] the user is not interested in the item, or that the user is not aware of its existence." While I agree with this statement, the Netflix dataset has ratings--why not report precision at k for that data?

Confidence in this Review

2-Confident (read it all; understood it all reasonably well)


Reviewer 4

Summary

This paper studies recommendation systems. The paper proposed a collaborative recurrent autoencoder (CRAE) which models the generation of content sequences under the collaborative filtering(CF) setting. The model generalizes the input of the recurrent deep learning to non-i.i.d. and provides a new denoising scheme and a learnable pooling scheme. Experiments on real data validates the performance of CRAE and CF.

Qualitative Assessment

The paper bridges the gap between the recurrent neural network (RNN) and CF which is a novel contribution in recommendation systems. By incorporating the CRAE, the order of the word is considered instead of bag of words only in traditional CNN. In addition, the paper extends the i.i.d input constraint of RNN to non-i.i.d one to adapt the CF setting by generalizing RNN to a hierarchical Bayesian model. Overall, I think the paper is technically sound and interesting. The only limitation I can see is that the performance gain shown in the experiments are marginal for recommendations. (relative boost 2.0% ~ 16.7% for CiteULike)

Confidence in this Review

1-Less confident (might not have understood significant parts)


Reviewer 5

Summary

This paper targets at an important problem in collaborative filtering - combining content information and user ratings to boost the accuracy of recommendations. Two RNNs are concatenated to form an encoder-decoder architecture to deal with the text sequence learning in recommender system, and this is the first work for using RNN for collaborative filtering. Different from LSTM that is prone to overfitting given insufficient data, this paper proposes to infuse (Gaussian) noise directly in the course of back and forward propagation, which is more simple and efficient than adding separate neural networks to take care of the noise distribution. In addition, two tricks are presented: i) wildcard based denoising -- semantical and random word replacement; ii) beta-pooling, which deals with the varying-length text sequence feature extraction task. The paper is well organized and the proposed method shows improvements in accuracy over state-of-the-art methods.

Qualitative Assessment

1) Some of the key arguments are not backed up with solid theoretical analysis or experimental studies. For instance, the authors proposed Robust Recurrent Networks to address the robustness problem of RNN, but there is no experiment showing how much better the proposed method can achieve in terms of robustness. The authors claimed that the proposed wildcard denoising can provide a relative accuracy boost of about 20% in CiteULike, but this argument is a little bit vague because the accuracy boost should be related to evaluation metrics and parameters, e.g., the M value in recall@M. Meanwhile, some details for how the wildcard denoising is performed are important but missing. Beta-pooling shall be compared with other alternative feature vector normalization techniques, the introduction for using this technique is not clearly presented in the context, which calls for clearance. Please also cite relevant papers if necessary. 2) The authors adopted averaged recall in the experiments. Ratings in Netflix dataset are not implicit, so that some ranking measures, e.g., NDCG, could be better than recall. Meanwhile, averaged recall is a biased measure because it treats users with different number of ratings equally. However, a few more correct recommendations on users with small number of ratings will cause larger increase on average recall than the same number of correct recommendations on users with large number of ratings, so that methods which perform better on users with small number of ratings will take some advantage. In such case, AUC could be a better choice. Overall, it will be more convincing if the authors could include some other evaluation metrics, e.g., NDCG or AUC as in [11]. 3) A minor issue in the experiment is that the M values in Recall@M start from 50. It is very important to see what the performance is before M=50. In many real-world problems, the number of recommendations should not be so large, and many users only receive a few recommendations at one time, e.g., 5-10 in Amazon.com. 4) The theoretical part of this paper is fair. Though this is the first work for adopting RNN in collaborative filtering, but the main actual contributions come from some aforementioned tricks, i.e. beta-pooling and wildcard denoising, which are not well motivated.

Confidence in this Review

2-Confident (read it all; understood it all reasonably well)


Reviewer 6

Summary

The authors propose a combination of an robust recurrent network (RNN) and collaborative filtering (CF). Over fitting is avoided using noise in the RNN and wildcard denosing. Wildcards are inserted into the training material at random places.

Qualitative Assessment

Majors: p. 5 Why do you use a slight asymmetry for the beta pooling, in general and in your experiments in particular? p. 6 The parameter P as different degrees of density needs further explanation! p. 7 How useful the results on the CiteULike dataset? The recall is smaller than 12% for M = 300. This is probably because there are only a few likes per author. Maybe there is a better dataset. Minors: p. 7 Labels in Fig 2 are barely readable. Also, label the panels. It only became apparent to me on p. 8 why there are four. What value has M in panel 1 and 2? p. 7 Did you implement the baseline algorithms yourself or were you able to use versions from the authors? Did the authors of the other methods process the same datasets? p. 8 It would be good to give a brief explanation or definition of the BLEU score (I know that you Ref. 12 probably has the definition). p. 7 What is the value for M in panel 1 and 2 of Fig 2? p. 8 (directly dele ting input...)

Confidence in this Review

1-Less confident (might not have understood significant parts)